# Negentropy Spectrum Decomposition and Its Application in Compound Fault Diagnosis of Rolling Bearing

**DOI:** 10.3390/e21050490

**Published:** 2019-05-13

**Authors:** Yonggang Xu, Junran Chen, Chaoyong Ma, Kun Zhang, Jinxin Cao

**Affiliations:** The Key Laboratory of Advanced Manufacturing Technology, Beijing University of Technology, Beijing 100124, China

**Keywords:** rolling bearing, compound fault, negentropy spectrum decomposition, fast empirical wavelet transform, spectral negentropy

## Abstract

The rolling bearings often suffer from compound fault in practice. Compared with single fault, compound fault contains multiple fault features that are coupled together and make it difficult to detect and extract all fault features by traditional methods such as Hilbert envelope demodulation, wavelet transform and empirical node decomposition (EMD). In order to realize the compound fault diagnosis of rolling bearings and improve the diagnostic accuracy, we developed negentropy spectrum decomposition (NSD), which is based on fast empirical wavelet transform (FEWT) and spectral negentropy, with cyclic extraction as the extraction method. The infogram is constructed by FEWT combined with spectral negentropy to select the best band center and bandwidth for band-pass filtering. The filtered signal is used as a new measured signal, and the fast empirical wavelet transform combined with spectral negentropy is used to filter the new measured signal again. This operation is repeated to achieve cyclic extraction, until the signal no longer contains obvious fault features. After obtaining the envelope of all extracted components, compound fault diagnosis of rolling bearings can be realized. The analysis of the simulation signal and the experimental signal shows that the method can realize the compound fault diagnosis of rolling bearings, which verifies the feasibility and effectiveness of the method. The method proposed in this paper can detect and extract all the fault features of compound fault completely, and it is more reliable for the diagnosis of compound fault. Therefore, the method has practical significance in rolling bearing compound fault diagnosis.

## 1. Introduction

Rolling bearings are among the most common and yet critical parts in rotating mechanical equipment. The working state of rolling bearings directly affects the performance of mechanical equipment. The fault of the rolling bearings is one of the main reasons leading to failures of mechanical equipment during operation. Bearing faults, which often occur gradually, represent one of the foremost causes of failures in the industry. Therefore, detection of their faults in the early stage is quite important to assure reliable and efficient operation [1]. Thus, it is of great significance to study the fault diagnosis technology of rolling bearings [2,3,4]. For the single fault diagnosis of rolling bearings, researchers have proposed Fourier transform [5], envelope analysis [6], empirical mode decomposition (EMD) [7], wavelet transform [8] and fast kurtogram [9], and they achieved good application results. In recent years, fault diagnosis methods based on intelligent classifier and machine learning have attracted the attention of scholars and made good progress [10,11]. Due to the complexity of equipment and the correlation of structures, engineering practice mightily indicates that faults emerged in rotating machinery usually shows, as the compound faults [12]. Dhamande et al. [13] proposed new compound fault features, extracted from the continuous and discrete wavelet transform of the vibration signal. It is found that these features have excellent diagnostic potential. This paper provides a new idea for composite fault diagnosis, but due to the limitations of the wavelet transform, it has not achieved ideal results. Yu et al. [14] proposed an improved morphological component analysis (MCA) method for the compound fault diagnosis of gearboxes. This method selects the optimal dictionary from the limited wavelet candidates, which restricts the generalizability of this method. Compound fault diagnosis is a difficult problem in the field of fault diagnosis, and it is also an urgent problem to be addressed [15].

Compound fault refers to failures in rotating machinery, where multiple faults occur simultaneously in rotating machinery and all the fault features are coupled together. Previous attempts to detect compound fault adopted intelligent classifiers, such as artificial neural network (ANN) [16], support vector machine (SVM) [17], and fuzzy inference [18]. Traditional machine learning methods require a large amount of labeled data to ensure the generalization of the algorithm. However, in practical problems, most of the samples are unlabeled, which results in unsatisfactory results of trained learning models in practical applications, and makes a large number of unlabeled samples useless [19]. In order to make compound fault diagnosis more efficient, decoupling technologies were developed. The measured signal is decomposed into sub-signals by wavelet transform or EMD, and the sub-signals are analyzed to diagnose compound fault. Xu et al. [20] proposed the compound fault diagnosis of rolling bearings based on dual-tree complex wavelet transform (DT-CWT). The compound fault signal is decomposed into several components of different frequency bands by DT-CWT, and ICA was used to separate the mixed signal, which consists of each component to eliminate the frequency aliasing, so that the detection and extraction of the fault features of compound fault are enabled. However, the wavelet transform is essentially a rigid method, corresponding to use prescribed wavelet basis function to process signals. So it cannot adaptively process a signal based on the information contained in the signal. Wang et al. [21] replaced wavelet transform with ensemble empirical mode decomposition (EEMD), and adaptively decomposed the compound fault signal into several intrinsic mode functions (IMF). The compound fault diagnosis can be realized by analyzing the IMF. However, due to the lack of sufficient theoretical support, the problems of modal aliasing, endpoint effect and over envelope in EMD method cannot be solved. Empirical wavelet transform (EWT), proposed by Gilles [22], can divide the signals adaptively and restrain modal aliasing to get more reasonable components. Kedadouche et al. [23] compared EWT with EMD and concluded that EWT is superior to EMD in mode estimates, and can reduce computation time. Yu et al. [24] applied EWT to compound fault diagnosis, and proved that the method has more attractive performance than wavelet transform and EMD. Scholars have studied that the EWT method causes modal aliasing due to the unreasonable boundary partition in the processing of noisy signals, which will increase the computational time and obtain more invalid components. Feng et al. [25] proposed a method for applying Adaptive Average Spectral Negentropy (AASN) to EWT analysis (AEWT), to accurately determine the optimal resonant demodulation frequency band, but this method cannot solve the problem of unreasonable boundary partition of EWT in essence. Xu et al. [26] proposed a novel method for selecting the more reasonable band to improve empirical wavelet transform. The fast empirical wavelet transform (FEWT) is proposed, replacing the boundary division method in EWT. Compared with EWT, FEWT can divide the frequency band more reasonably and extract the effective empirical modes, and improve the operation speed. Xu et al. used FEWT combined with kurtosis to construct kurtogram, and achieved good results. Since experiments show that this method is not effective in the presence of impulsive noise.

In fault diagnosis, the main challenge is to find the most suitable frequency band for demodulation. Antoni [9] proposed spectral kurtosis (SK) to solve this problem. The SK is a statistical tool which can indicate the presence of a series of transients and their locations in the frequency domain. Antoni et al. applied SK to vibration monitoring of rotating machinery and proposed the concept of kurtogram [27]. Additionally, the proposal of fast kurtogram [28] makes up for the shortcoming of kurtogram. However, fast kurtogram often fails in the presence of impulsive noise, as the SK cannot recognize whether a series of transients are repetitive or not. In order to remedy this defect, Antoni [29] introduced the concept of spectral negentropy, and used the squared envelope (SE) to characterize the pulse features caused by repetitive faults, and the squared envelope spectrum (SES) to characterize the cyclostationality. Spectral negentropy can characterize the periodicity and cyclostationality of the signal, as well as accurately extract the fault feature of the signal in the presence of impulsive noise [30,31]. The method based on FEWT and spectral negentropy can accurately extract fault features. However, due to the complexity of compound fault signals, this method cannot diagnose compound fault.

In this paper, a compound fault diagnosis method based on negentropy spectrum decomposition (NSD) is proposed. By presupposing the initial reconstructed points and accumulating them, a series of empirical modes are obtained by reconstructing the key functions. Infogram is constructed by combining spectral negentropy. The filtered signal is used as a new measured signal, and the above operation is repeated until the signal no longer contains obvious fault feature information. The paper is organized as follows. Section 2 introduces FEWT and its combination with SK; Section 3 proposes the NSD method and its application in the compound fault diagnosis of rolling bearings; Section 4 verifies that the method can diagnose compound fault of rolling bearings through simulation signals; Section 5 verifies the effectiveness of the method through experiments.

## 2. Exposition of Fast Empirical Wavelet Transform

### 2.1. FEWT Method

Gilles proposed EWT based on EMD and wavelet analysis. FFT is used to get the spectrum. The spectrum is normalized to [0, π], then divided into N continuous intervals, and N−1 boundaries are obtained. With the boundary of the interval as the center, the transition phase is defined to establish the window base.

According to the definition of empirical wavelet, it is necessary to construct an appropriate band-pass filter for each frequency band to extract the corresponding signals for signal reconstruction. Gilles constructed empirical wavelet according to the Meyer wavelet construction method. In the transitional section of two boundaries in the frequency band, a set of orthogonal triangular functions are designed, and an invariant constant is set within the frequency band. The empirical scaling function ${\hat{\phi }}_{n}\left(\omega \right)$ and the empirical wavelets ${\hat{\psi }}_{n}\left(\omega \right)$ can be respectively defined as:(1)$${\hat{\emptyset }}_{n}\left(\omega \right)=(\begin{array}{c} 1;\left|\omega \right|\leq \left(1-\gamma \right)\omega_{n}\\ \cos\left[\frac{\pi }{2}\beta \left(\frac{1}{2{\gamma \omega }_{n}}\left(\left|\omega \right|-\left(1-\gamma \right)\omega_{n}\right)\right)\right]\\ 0;\mathrm{others} \end{array};\left(1-\gamma \right)\omega_{n}\leq \left|\omega \right|\leq \left(1+\gamma \right)\omega_{n}$$
(2)$${\hat{\psi }}_{n}\left(\omega \right)=(\begin{array}{c} 1;\left(1+\gamma \right)\omega_{n}\leq \left|\omega \right|\leq \left(1-\gamma \right)\omega_{n+1}\\ \cos\left[\frac{\pi }{2}\beta \left(\frac{1}{2{\gamma \omega }_{n+1}}\left(\left|\omega \right|-\left(1-\gamma \right)\omega_{n+1}\right)\right)\right];\left(1-\gamma \right)\omega_{n+1}\leq \left|\omega \right|\leq \left(1+\gamma \right)\omega_{n}\\ \sin\left[\frac{\pi }{2}\beta \left(\frac{1}{2{\gamma \omega }_{n}}\left(\left|\omega \right|-\left(1-\gamma \right)\omega_{n+1}\right)\right)\right];\;\left(1-\gamma \right)\omega_{n}\leq \left|\omega \right|\leq \left(1+\gamma \right)\omega_{n}\;\\ 0;\mathrm{others} \end{array}$$
where the transition function β(x), the coefficient γ, and the transition phase $\tau_{n}$ are:(3)$$\beta \left(x\right)=x^{4}\left(35-84x+70x^{2}-20x^{3}\right)$$
(4)$$\gamma <min\left(\frac{\omega_{n+1}-\omega_{n}}{\omega_{n+1}+\omega_{n}}\right)$$
(5)$$\tau_{n}={\gamma \omega }_{n},0<\gamma <1$$

The detail coefficients of EWT is expressed as:(6)$$W_{f}^{\varepsilon }\left(n,t\right)=\;<f\left(t\right),\Psi_{n}\left(t\right)>\;=\int f\left(\tau \right)\bar{\Psi_{n}\left(\tau -t\right)}d\tau =F^{-1}\left(\hat{f}\left(\omega \right)\bar{{\hat{\Psi }}_{n}\left(\omega \right)}\right)$$

In which, F(·) is the Fourier transform and $F^{-1}\left(\cdot \right)$ is the inverse Fourier transform. The approximation coefficients is expressed as:(7)$$W_{f}^{\varepsilon }\left(0,t\right)=\;<f\left(t\right),\emptyset_{1}\left(t\right)>\;=\int f\left(\tau \right)\bar{\emptyset_{1}\left(\tau -t\right)}d\tau =F^{-1}\left(\hat{f}\left(\omega \right)\bar{{\hat{\emptyset }}_{1}\left(\omega \right)}\right)$$

The signal f can be reconstructed by:(8)$$f\left(t\right)=W_{f}^{\varepsilon }\left(0,t\right)*\phi_{1}\left(t\right)+\overset{N}{\underset{n=1}{\sum }}W_{f}^{\varepsilon }\left(0,t\right)*\psi_{n}\left(t\right)\;=F^{-1}\left({\hat{W}}_{f}^{\varepsilon }\left(0,\omega \right){\hat{\phi }}_{1}\left(\omega \right)+\overset{N}{\underset{n=1}{\sum }}{\hat{W}}_{f}^{\varepsilon }\left(n,\omega \right)*{\hat{\psi }}_{n}\left(\omega \right)\right)$$

The empirical modes can be obtained by:(9)$$\{\begin{array}{c} f_{0}\left(t\right)=W_{f}^{\varepsilon }\left(0,t\right)*\phi_{1}\left(t\right)\\ f_{k}\left(t\right)=W_{f}^{\varepsilon }\left(k,t\right)*\psi_{k}\left(t\right) \end{array}$$

After obtaining the empirical mode of the signal, the Hilbert envelope method is used to calculate the instantaneous frequency, and then the fault diagnosis can be realized.

The disadvantage of EWT is that while a large number of parameters are set to divide the boundary adaptively, an appropriate boundary cannot be obtained in most cases. FEWT proposes a new boundary partitioning method to replace the boundary partitioning method in EWT. Specific steps of FEWT are described in Reference [26]:

Step 1: The Fast Fourier Transform (FFT) is used to obtain the spectrum Y(f) of the measured signal Y(t). The key function K(f) is obtained by FFT again.

Step 2: A part of the key function is extracted, the inverse Fourier transform is performed and the trend component in the spectrum is obtained. The degree of trend is related to the key function extracted.

Step 3: The wavelet threshold method is used to optimize the trend, reduce the noise interference, and get more reasonable boundaries.

Step 4: Empirical wavelet is constructed to decompose and reconstruct signals.

A simulation signal is constructed to verify that FEWT can divide the boundary more reasonably. First, a series of transient are constructed to simulate the fault signals of bearing outer rings and the inherent frequency of the signals is set as $f_{n1}=2000\;\mathrm{Hz}$, the damping coefficient is set as g=0.08, and the repeated cycle is set as T=0.01 s. So the characteristic frequency is 100 Hz. Second, the non-impact and non-fault modulation signals $s_{2}$ in rotating machinery operation are simulated: the inherent frequency of the signal is $f_{n2}=5000\;\mathrm{Hz}$, the width of the side band is 100 Hz, and the noise is η=SNR(−0.03 dB).
(10)$$\left\{ \begin{array}{l} s_{1}=4e^{-g\times 2{\pi f}_{n1}t}\times \sin\left(2{\pi f}_{n1}t\times \sqrt{1-g^{2}}\right)\\ s_{2}=\sin\left(100\pi t\right)\sin\;\left(2{\pi f}_{n2}t+\sin\left(100\pi t\right)\right)+0.5\cos\left(4\pi t\right)\cos\left(100\pi t\right)\\ s\;=s_{1}+s_{2}+\eta \end{array}\right.$$

Figure 1a,b show the components of the signal. The signal-to-noise ratio signal is shown in Figure 1c and its spectrum in Figure 1d. Two methods are used to process signals and the following results can be obtained. Figure 2a is the result of EWT. As can be seen in part A, the EWT method incorrectly divides the resonance frequency band of the 2000 Hz pulse signal; part B divides up too many invalid components. The shortcomings of the above EWT partitioning methods negatively affect the analysis of components and increase the calculation time. Figure 2b is the result of FEWT. Compared with EWT by results, FEWT can find more reasonable boundaries.

### 2.2. FEWT Combined with Kurtogram

In the fast kurtogram proposed by Antoni [27], the SK is calculated after decomposing the signal using the 1/3-binary tree filter bank. Reference [26] draws on the fast kurtogram to obtain a new kurtogram based on FEWT. This method can set up the calculation speed and find the position of periodic shock in the frequency domain. Specific steps of the method are described in Reference [26]:

Step 1: Set the initial number of reconstructed points to five and named ResCot.

Step 2: A series of signal components can be obtained by FEWT with ResCot as the reconstructed points.

Step 3: If ResCot<60, set ResCot=ResCot+5 and repeat step 2 until ResCot>60. Kurtogram is constructed by calculating the SK of all signal components.

Step 4: Extract the components with the maximum kurtosis in the kurtogram; calculate envelope spectrum and extract faults.

Reference [26] shows that the method based on FEWT and kurtogram has certain advantages over fast kurtogram. However, this method has its shortcomings: when there is impulse noise interference, impulse noise will conceal fault features in the kurtogram. The following example shows the shortcomings of SK in the presence of impulsive noise. A series of transients is synthesized according to the model of Reference [32] and the inherent frequency of the series of transients is 100 Hz, the damping coefficient is g=0.08. Adding impulsive noise at t=1, the inherent frequency of the impulsive noise is 300 Hz, the damping coefficient is g=0.02. Figure 3a shows a series of transients produced by a faulty rolling element bearing and further corrupted by impulsive noise at time instant t=1. Figure 3b shows the SNR signal. The method based on FEWT and kurtogram is used to process the signal in Figure 3b, and the kurtogram obtained is shown in Figure 4a. It can be seen that the impulsive noise with an inherent frequency of 300 Hz dominates the kurtogram, and conceals a series of transients with inherent frequency of 100 Hz.

In addition, when the method based on FEWT and kurtogram is used to process compound fault signals, only obvious fault features can be found in the signals by extracting a single feature component for analysis, but other faults cannot be identified. The following example shows that the method cannot diagnose and identify the compound fault of rolling bearings. According to the model of References [33,34], the compound fault signals of inner and outer rings of rolling bearings are simulated. The simulation signals of rolling bearing outer ring fault are constructed as:(11)$$\{\begin{array}{c} x_{o}\left(t\right)=s\left(t\right)+n\left(t\right)=\sum_{i}A_{i}H\left(t-{\mathrm{iT}}_{1}-\tau_{i}\right)+n\left(t\right)\\ A_{i}=\mathrm{constant}\\ H\left(t\right)=\exp\left(-\mathrm{gt}\right)\cos\left(2\pi f_{n1}t+\phi_{\omega }\right) \end{array}$$

The simulation signals of rolling bearing inner ring fault are constructed as:(12)$$\{\begin{array}{c} x_{i}\left(t\right)=s\left(t\right)+n\left(t\right)=\sum_{i}A_{i}H\left(t-{\mathrm{iT}}_{2}-\tau_{i}\right)+n\left(t\right)\\ A_{i}=A_{0}\cos2\left(2\pi f_{r}t+\phi_{A}\right)\\ H\left(t\right)=\exp\left(-\mathrm{gt}\right)\cos\left(2\pi f_{n2}t+\phi_{\omega }\right) \end{array}$$

Simulated signals of compound fault can be obtained from:(13)$$x\left(t\right)=x_{o}\left(t\right)+x_{i}\left(t\right)$$
where n(t)=SNR, g is the damping coefficient, and $\tau_{i}$ represents slight fluctuations in the impact process. The inherent frequency of the outer ring is $f_{n1}=1000\;\mathrm{Hz}$, the fault characteristic frequency of outer ring is $f_{\mathrm{out}}=\frac{1}{T_{1}}=100\;\mathrm{Hz}$; The inherent frequency of the inner ring is $f_{n2}=5000\;\mathrm{Hz}$, the fault characteristic frequency of inner ring is $f_{\mathrm{in}}=\frac{1}{T_{2}}=150\;\mathrm{Hz}$. The rotation frequency is $f_{r}=15\;\mathrm{Hz}$. Figure 5a shows the simulated signal of the outer ring fault, Figure 5b shows the simulated signal of inner ring fault, Figure 5c shows the compound fault signal, and Figure 5d shows the SNR signal. The method based on FEWT and kurtogram is used to process the signal in Figure 5d, the kurtogram is shown in Figure 6a. When the CotPoint=45, the maximum component of kurtosis appears. Its kurtosis is $K_{\max}=16.39$, the center frequency is $f_{c}=4985.7939\;\mathrm{Hz}$; the boundaries are [4664.8 Hz;5306.8 Hz], the bandwidth is Bw=642 Hz. Figure 6b shows the extracted signal and its envelope spectrum. There are obvious characteristic frequencies and frequency doubling in the envelope spectrum, which are similar to the fault characteristic frequency of the inner ring, but the fault characteristic of the outer ring cannot be found in envelope spectrum. Therefore, the method based on FEWT and kurtogram cannot identify the compound fault signal.

## 3. Proposed Method of Fast Negentropy Spectrum Decomposition

Rolling bearings are among the most common and yet critical parts in rotating mechanical equipment, so the research on the fault diagnosis technology of rolling bearings has far-reaching significance. The single fault characteristics of the outer ring, inner ring and rolling body of rolling bearings are similar to those of the unilateral oscillation attenuation waveform, while the compound fault features are the coupling of two or more single fault features, so the single fault diagnosis method cannot diagnose the compound fault. In this section, combined with FEWT and spectral negentropy, a compound fault diagnosis method based on NSD is proposed.

### 3.1. Exposition of Spectral Negentropy

Entropy is a multifunctional concept for measuring system disorders. The appearance of fault impulse means that the equilibrium state of the system is broken and the entropy of the system has been changed. The spectral entropy is defined as the entropy in the signal frequency band. When the bearing is in normal condition, the energy fluctuation of the signal is constant and the spectral entropy value is the maximum. On the other hand, when the energy fluctuation caused by fault pulse changes, the spectral entropy value is the minimum, which is contrary to the change of kurtosis index. In order to let it have the same physical meaning as spectral kurtosis, the negative value of spectral entropy is defined as spectral negentropy.

Formally, consider a discrete-time signal x(n),n=0,…,L−1, of length L, and the square envelope of its frequency band $\left[f-\frac{\Delta f}{2};f+\frac{\Delta f}{2}\right]$ is
SE_X_ = |x(n; f, Δf) + jH(x(n; f, Δf))|^2^(14)
where H(·) is a Hilbert transformation. Spectral negentropy in time domain reads
(15)$$\Delta I_{e}\left(f;\Delta f\right)=\langle \frac{{\mathrm{SE}}_{x}{\left(n;f,\Delta f\right)}^{2}}{\langle {\mathrm{SE}}_{x}{\left(n;f,\Delta f\right)}^{2}\rangle }\rangle \ln\langle \frac{{\mathrm{SE}}_{x}{\left(n;f,\Delta f\right)}^{2}}{\langle {\mathrm{SE}}_{x}{\left(n;f,\Delta f\right)}^{2}\rangle }\rangle$$
where 〈·〉 is the mean operation. Similar to SK, spectral negentropy in time domain can characterize the pulse feature caused by repetitive faults. Note that repetitive faults are not periodic in general, but rather cyclostationary, and they can be defined by spectral negentropy in the frequency domain
(16)$$\Delta I_{E}\left(f;\Delta f\right)=\langle \frac{{\left|{\mathrm{SES}}_{x}\left(\alpha ;f,\Delta f\right)\right|}^{2}}{\langle {\left|{\mathrm{SES}}_{x}\left(\alpha ;f,\Delta f\right)\right|}^{2}\rangle }\rangle \ln\left(\frac{{\left|{\mathrm{SES}}_{x}\left(\alpha ;f,\Delta f\right)\right|}^{2}}{\langle {\left|{\mathrm{SES}}_{x}\left(\alpha ;f,\Delta f\right)\right|}^{2}\rangle }\right)$$

In which, ${\left|{\mathrm{SES}}_{x}\left(\alpha ;f,\Delta f\right)\right|}^{2}$ is the square envelope spectrum in the frequency band
(17)$${\mathrm{SES}}_{x}\left(\alpha ;f,\Delta f\right)=F\left({\mathrm{SE}}_{x}\left(n;f,\Delta f\right)\right)$$
where F(·) is the Fourier transform, and α is the change of the frequency. The weighted average of $\Delta I_{e}\left(f;\Delta f\right)$ and $\Delta I_{e}\left(f;\Delta f\right)$ is used to obtain the average spectral negentropy, and the periodicity and cyclostationality of the signal are measured simultaneously. Mean spectral negativity is seen that
(18)$$\Delta I_{1/2}\left(f;\Delta f\right)=\frac{\Delta I_{e}\left(f;\Delta f\right)}{2}+\frac{\Delta I_{E}\left(f;\Delta f\right)}{2}$$

Compared with spectral kurtosis, spectral negentropy can identify fault features in the presence of impulsive noise. Similar to the combination of FEWT and spectral kurtosis, a method based on FEWT and spectral negentropy is used to analyze the signal in Figure 3b, and the infogram is displayed in Figure 4b–d. It can be seen that the impulsive noise at frequency 300 Hz dominates the SE infogram. However, SES infogram and average infogram show radically different behavior: it is insensitive to the impulsive noise and clearly reveals the series of transients at 100 Hz. Therefore, the method based on FEWT and spectral negentropy can identify fault features in the presence of impulsive noise.

### 3.2. Negentropy Spectrum Decomposition Method

When compound faults occur in rolling bearings, they are coupled with each other and have their own resonance frequency. The resonance frequencies of faults are often different, that is, the resonance frequency bands exist in different frequency bands. By filtering based on FEWT and spectral negentropy, obvious fault features can be extracted. However, the extracted frequency band often does not contain other fault features. The compound fault simulation signal in Figure 7a shows this situation, and its spectrum is shown in Figure 7b. The inherent frequency of the outer ring is $f_{n1}=1000\;\mathrm{Hz}$, and the inherent frequency of the inner ring is $f_{n2}=6000\;\mathrm{Hz}$. The infogram is shown in Figure 8a. When the CotPoint=60, the maximum component of spectral negentropy appears. Its spectral negentropy is ${\mathrm{Info}}_{\max}=3.23$, the centre frequency is $f_{c}=5995.7774\;\mathrm{Hz}$; the boundaries are [5790.3 Hz;6201.3 Hz], the bandwidth is Bw=411 Hz. Figure 8b shows the extracted signal and its envelope spectrum, and only the inner fault features can be seen in the envelope spectrum.

In another case, when the inherent frequencies of the two faults are close to each other, the extracted components contain a small number of features of secondary faults. Therefore, weak characteristic frequency of secondary faults can be seen in the envelope spectrum, but they are seriously disturbed by noise and cannot be used as a basis for judgment. The compound fault simulation signal in Figure 7c shows this situation, and its spectrum is shown in Figure 7d. The inherent frequency of the outer ring is $f_{n1}=5000\;\mathrm{Hz}$, and the inherent frequency of the inner ring is $f_{n2}=6000\;\mathrm{Hz}$. The infogram is shown in Figure 9a. When the CotPoint=55, the maximum component of spectral negentropy appears. Its spectral negentropy is ${\mathrm{Info}}_{\max}=3.38$, the center frequency is $f_{c}=5938.1534\;\mathrm{Hz}$; the boundaries are [5563.7 Hz;6312.7 Hz], the bandwidth is Bw=749 Hz. Figure 9b shows the extracted signal and its envelope spectrum. In the envelope spectrum, the inner ring fault features are obvious, while the outer ring fault features are seriously disturbed by noise and therefore are not obvious. Thus, the method based on FEWT and infogram cannot diagnose compound fault.

A novel method of compound fault diagnosis based on NSD is proposed to solve the problem that extracting a single component cannot diagnose compound fault. FEWT is used to decompose the signal adaptively, and combining the spectral negentropy to construct the infogram, the frequency band with the maximum spectral negentropy in the infogram can be extracted. After the extraction of the frequency band, the signal repeats the above operation and extracts the characteristic components cyclically, which can diagnose compound fault. Figure 10 shows the process of the proposed method. The specific steps are as follows:

Step 1: A series of boundaries in frequency domain can be obtained by presupposing initial reconstruction points and accumulating them. Empirical wavelet and adaptive filter are constructed to decompose the signal into a series of components.

Step 2: The spectral negentropy of each component is calculated and the infogram is constructed to determine the parameters of the band-pass filter.

Step 3: Extract the components with the maximum spectral negentropy in the infogram; calculate envelope spectrum and extract faults.

Step 4: The signal after component extraction is used as a new measured signal to determine whether it contains obvious faults. If it does, return to the first step; else, proceed to the next step.

Step 5: All the components extracted are analyzed to diagnose compound fault.

### 3.3. Cyclic Cutoff Conditions

The NSD method extracts a series of transients in turn, and stops extraction when there are no obvious fault features in the signal. It can be considered that the extracted signal contains only noise, modulation and a series of residual transients. In order to explore the spectral negentropy of noise with different SNR, modulated signals with different frequencies and a series of transients, the following simulation signals are constructed and their spectral negentropy is calculated.

(A) $s_{1}=\eta$, SNR=[−20, −15,−10,−5,−3,−2,−1,0], the spectral negentropy of noise with different SNR is calculated.

(B) $s_{2}=2\cos\left(2\pi \mathrm{ft}\right)$, the spectral negentropy of cosine signals with different frequencies is calculated.

(C)
(19)$$\{\begin{array}{l} s_{3}=2e^{-g\times 2{\pi f}_{1}t}\times \sin\left(2\pi f_{1}t\times \sqrt{1-g^{2}}\right)\\ s_{4}=2e^{-g\times 2{\pi f}_{2}t}\times \sin\left(2\pi f_{2}t\times \sqrt{1-g^{2}}\right)\\ s_{5}=2e^{-g\times 2{\pi f}_{3}t}\times \sin\left(2\pi f_{3}t\times \sqrt{1-g^{2}}\right) \end{array}$$

C=[60, 70,80,90,100,110,120,130], repetition period T=1/C, $f_{1}=1000\;\mathrm{Hz}$, $f_{2}=2000\;\mathrm{Hz}$, $f_{3}=3000\;\mathrm{Hz}$, the spectral negentropy of a series of transient with inherent frequencies of 1000 Hz, 2000 Hz and 3000 Hz are calculated for different repetition periods.

Figure 11 shows the results of the experiment. The blue line is the spectral negentropy of noise signals with different SNR. The fluctuation of the line is very small. It can be seen that noise has little effect on the spectral negentropy. The black line is the spectral negentropy of cosine signals with different frequencies, and the line is close to a straight line. It can be seen that the frequency of modulated signal has little effect on the spectral negentropy. Green, cyan and purple lines show the spectral negentropy of a series of transient with inherent frequencies of 1000 Hz, 2000 Hz and 3000 Hz at different repetition periods, respectively. It can be seen that when the inherent frequency is unchanged, the greater the repetition period, the greater the spectral negentropy; when the repetition period is unchanged, the greater the inherent frequency, the greater the spectral negentropy.

Spectral negentropy of the modulated signal with Gaussian noise is determined by a large number of experiments. The simulation signal $s=s_{1}+s_{2}$ is constructed, the SNR and cosine frequency are randomly selected from a certain range, SNR=[−20;0], f=[10;300]. A thousand simulation signals are randomly constructed and the spectral negentropy is calculated. Figure 12 shows the distribution range of spectral negentropy. Spectral negentropy is mainly concentrated in the range of [0.7;0.75]. The maximum and minimum spectral negentropy are 0.8083 and 0.5281, respectively. The number of spectral negentropy greater than 0.8 in all simulation signals is 3. A series of residual transients in the signal that no longer have periodicity and have little influence on the spectral negentropy. Therefore, it can be determined that when the spectral negentropy is less than 0.8, the signal no longer contains obvious fault characteristics. In the NSD method, the cyclic cutoff condition is set to the spectral negentropy less than 0.8, which can extract all the fault features in the signal, avoid excessive cycles and improve the diagnostic efficiency.

## 4. Simulated Signal Verification

The compound fault simulated signal in Figure 5d is decomposed by FEWT, and a series of components is obtained. The spectral negentropy of each component is calculated and the infogram is constructed, as shown in Figure 13a. When the CotPoint=60, the maximum component of spectral negentropy appears. Its Spectral negentropy is ${\mathrm{Info}}_{\max}=4.04$, the centre frequency is $f_{c}=996.3588\mathrm{Hz}$, close to outer ring’s inherent frequency of $f_{n1}=1000\;\mathrm{Hz}$; the boundaries are [852.4 Hz;1140.4 Hz], the bandwidth is Bw=288 Hz. Figure 13b shows the extracted signal C1 and its envelope spectrum. The characteristic frequency of the outer ring fault can be seen in the envelope spectrum. This suggests the existence of the outer race bearing fault.

C1 is extracted from the measured signal and a new measured signal is obtained. Figure 14a shows the measured signal and Figure 14b is the spectrum of a. Figure 14c shows the new measured signal and Figure 14d is the spectrum of c. It can be seen from the spectrum of the new measured signal that the fault features of the outer ring have been extracted, leaving only the residual resonance frequency band. The new measured signal is decomposed by FEWT, and a series of components are obtained. Spectral negentropy of each component is calculated and infogram is constructed, as shown in Figure 15a. When the CotPoint=25, the maximum component of spectral negentropy appears. Its Spectral negentropy is ${\mathrm{Info}}_{\max}=3.25$, the centre frequency is $f_{c}=4951.7944\;\mathrm{Hz}$, close to inner ring’s inherent frequency of $f_{n2}=5000\;\mathrm{Hz}$; the boundaries are [4641.3 Hz;5262.3 Hz], the bandwidth is Bw=621 Hz. In Figure 15b, the characteristic frequency of 150Hz of the simulated signal is extracted effectively, and its sideband and frequency doubling of 300Hz are also very obvious. This suggests the existence of an inner race bearing fault. It can be concluded that the bearing has a compound fault of inner and outer rings. That is, the proposed method can avoid misdiagnosis, and can separate all compound fault characteristic frequencies of the simulated signal x(t).

## 5. Applications

This experiment uses the fault bearing test data collected by the laboratory at Xi’an Jiaotong University. The Spectra Quest, Inc test bench is shown in Figure 16. The actual motor frequency is 33.6 Hz, the motor speed is 1450 r/min, and the sampling frequency is Fs=12,000 Hz. The sensor is connected to the outer ring of the motor end bearing. After the calculation, the fault characteristic frequency of the inner ring of the bearing can be obtained as $f_{\mathrm{in}}=165.03\;\mathrm{Hz}$, the fault characteristic frequency of the outer ring of the bearing can be obtained as $f_{\mathrm{out}}=102.22\mathrm{Hz}$

Figure 17a shows the measured signal and its spectrum, Figure 17b shows the envelope of the signal, the type of bearing failure cannot be determined. The measured signal is decomposed by FEWT, and a series of components is obtained. The spectral negentropy of each component is calculated and the infogram is constructed, as shown in Figure 18a. When the CotPoint=10, the maximum component of spectral negentropy appears. Its Spectral negentropy is ${\mathrm{Info}}_{\max}=1.76$, the centre frequency is $f_{c}=5172.5\;\mathrm{Hz}$; the boundaries are [4345 Hz;6000 Hz], the bandwidth is Bw=1655 Hz. Figure 18b shows the extracted signal C1 and its envelope spectrum. As shown in Figure 18b, there are obvious peak values at the fault characteristic frequency $f_{\mathrm{out}}=102.22\;\mathrm{Hz}$ of the outer race and its harmonics (i.e., 204.44 Hz), and the spectra are extremely explicit. This suggests the existence of the outer race bearing fault.

C1 is extracted from the measured signal and a new measured signal is obtained. Figure 19a shows the new measured signal after C1 extraction and its spectrum. In the spectrum, there is no frequency component of the frequency band $f_{C1}=\left(4345\;\mathrm{Hz},6000\;\mathrm{Hz}\right)$. Figure 19b shows the envelope of the signal. The new measured signal is decomposed by FEWT, and a series of components are obtained. The spectral negentropy of each component is calculated and infogram is constructed, as shown in Figure 20a. When the CotPoint=15, the maximum component of spectral negentropy appears. Its Spectral negentropy is ${\mathrm{Info}}_{\max}=1.28$, the centre frequency is $f_{c}=3806.6667\;\mathrm{Hz}$; the boundaries are [2806.2 Hz;4753.2 Hz], the bandwidth is Bw=1893 Hz. Figure 20b shows the extracted signal C2 and its envelope spectrum. The characteristic frequency of inner ring fault can be seen in the envelope spectrum. This suggests the existence of the inner race bearing fault.

C2 is extracted from the measured signal and a new measured signal is obtained. Figure 21a shows the new measured signal after C2’s extraction and its spectrum. In the spectrum, there is hardly any frequency component of the frequency band (2806.2 Hz,6000 Hz). Figure 19b shows the envelope of the signal. The spectral negentropy of the new measured signal is 0.76, so the cyclic extraction is stopped. It can be concluded that the bearing has a compound fault of inner and outer rings, which is consistent with the simulated bearing fault. Thus, the NSD method can effectively diagnose compound fault of rolling bearings.

## 6. Conclusions

In practice, the faults of rotating machinery are usually compound fault. Compound fault contains multiple fault features and all the fault features are coupled together, making it difficult to diagnose compound fault. Therefore, exploring a reliable compound fault diagnosis method is one of the most urgent problems in the field of mechanical condition monitoring and fault diagnosis (CMFD). To overcome this challenge, a novel compound fault diagnosis method based on NSD is proposed in this paper. Combining FEWT and spectral negentropy to construct the infogram, the fault feature information in the signal can be extracted accurately in the presence of impulsive noise. The filtered signal is used as a new measured signal, and the above operation is repeated until the signal no longer contains obvious fault feature information. All fault features contained in the signal can be extracted sequentially. Compound fault diagnosis can be realized by envelope demodulation.

The method applies to the compound fault diagnosis of rolling bearings, and the feasibility and effectiveness of the method are verified. Compared with EWT, FEWT can divide more reasonable boundaries and has better adaptability. Compared with spectral kurtosis, spectral negentropy can extract fault features more accurate in the presence of impulsive noise. When it is used to detect whether there are obvious faults in the signal, the spectral negentropy is more stable and reliable. Therefore, the method based on FEWT and spectral negentropy can accurately extract fault features in signals. Through the extraction method of cyclic extraction, all the fault features in the composite fault signal can be extracted in turn, thus ensuring the accuracy of the compound fault diagnosis.

## Figures and Tables

**Figure 1 entropy-21-00490-f001:**
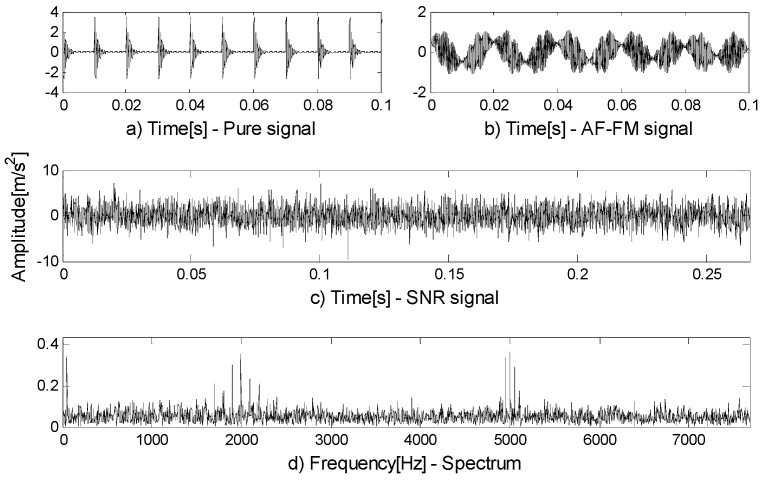
(**a**) The pure signal; (**b**) the amplitude-modulated and frequency-modulated (AM-FM) signal; (**c**) the signal-to-noise ratio (SNR) signal; (**d**) the spectrum of (**c**).

**Figure 2 entropy-21-00490-f002:**
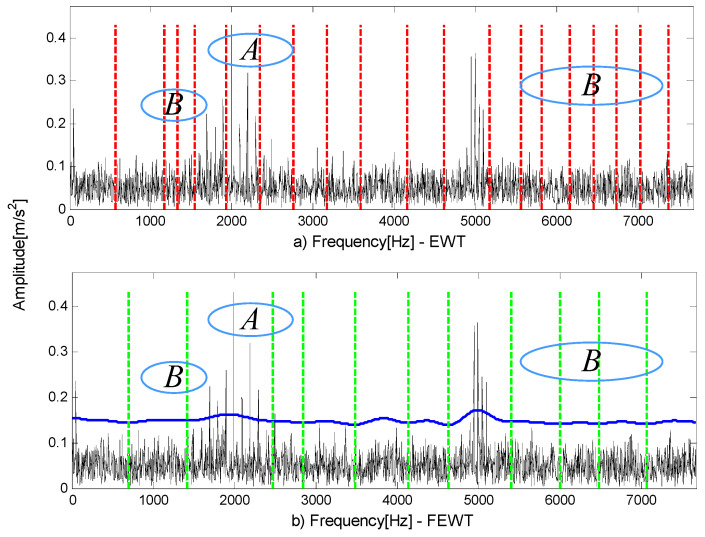
The boundaries divided by (**a**) the empirical wavelet transform (EWT) method and (**b**) the fast empirical wavelet transform (FEWT) method (purple line: trend component).

**Figure 3 entropy-21-00490-f003:**
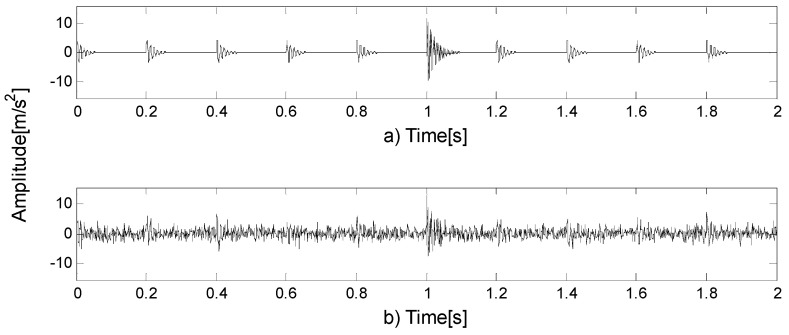
(**a**) A series of transients further corrupted by impulsive noise; (**b**) the SNR signal.

**Figure 4 entropy-21-00490-f004:**
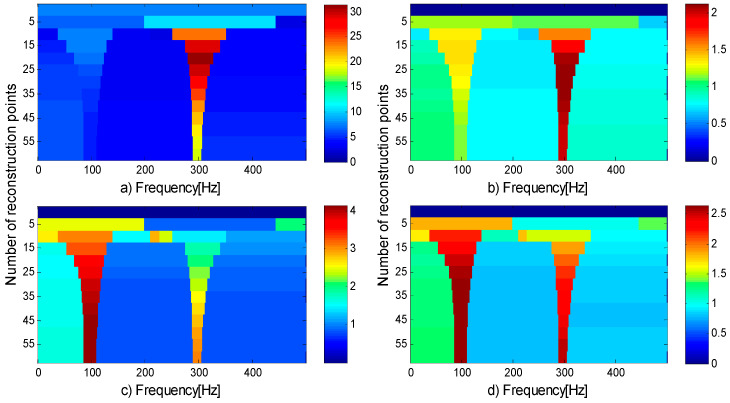
(**a**) Kurtogram; (**b**) SE infogram; (**c**) SES infogram; (**d**) average infogram of the signal in Figure 3b.

**Figure 5 entropy-21-00490-f005:**
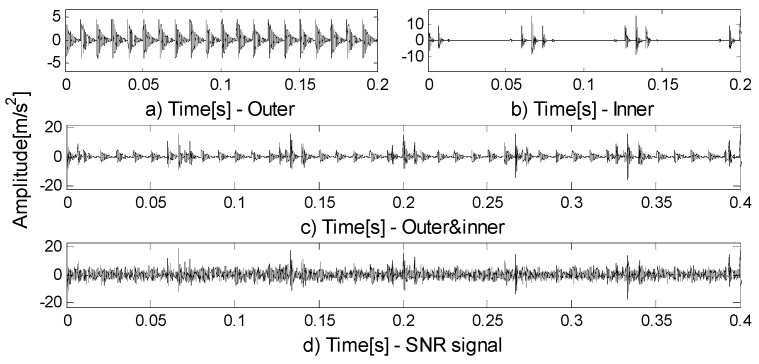
(**a**) The simulated signal of outer ring fault; (**b**) the simulated signal of inner ring fault; (**c**) the compound fault signal; (**d**) the SNR signal.

**Figure 6 entropy-21-00490-f006:**
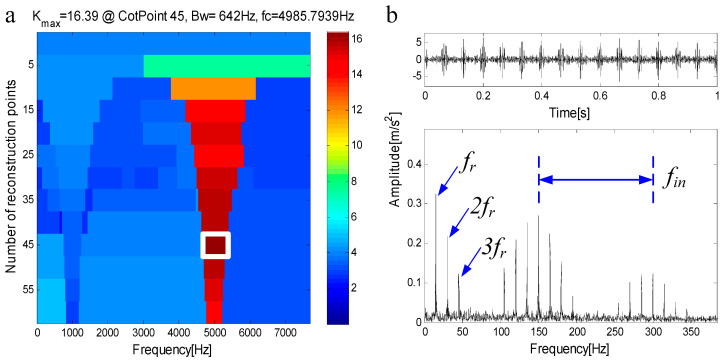
(**a**) Kurtogram based on FEWT; (**b**) component with ResPoint=45 and its envelope.

**Figure 7 entropy-21-00490-f007:**
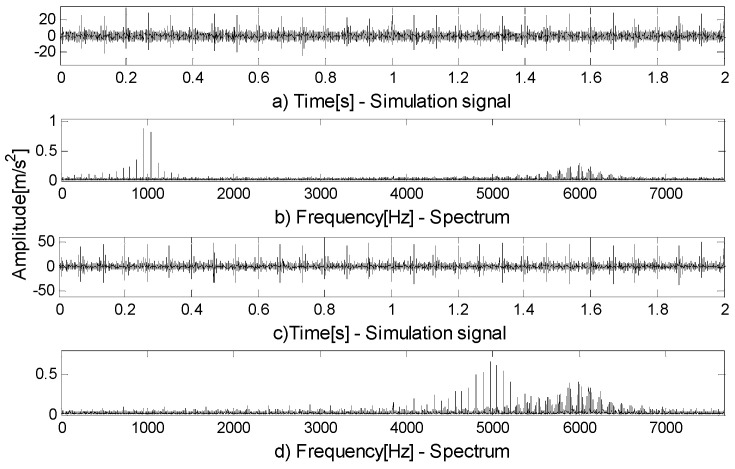
(**a**) Signal in the first case; (**b**) spectrum of (**a**); (**c**) signal in the second case; (**d**) spectrum of (**a**).

**Figure 8 entropy-21-00490-f008:**
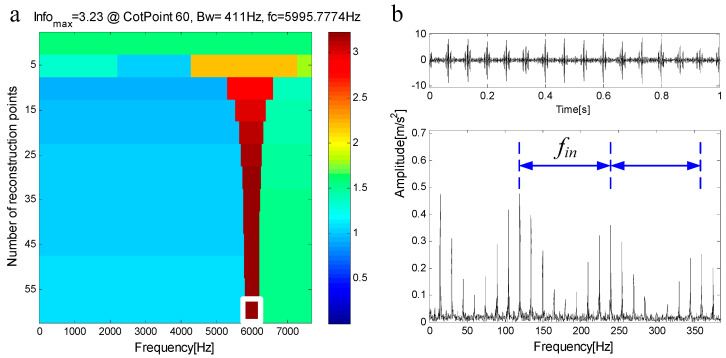
(**a**) Infogram based on FEWT; (**b**) component with ResPoint=60 and its envelope.

**Figure 9 entropy-21-00490-f009:**
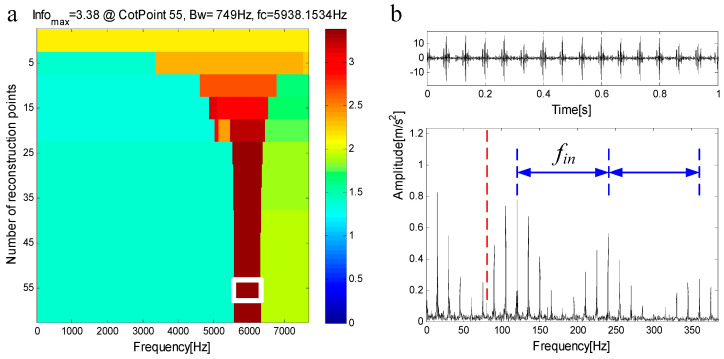
(**a**) Infogram based on FEWT; (**b**) component with ResPoint=55 and its envelope.

**Figure 10 entropy-21-00490-f010:**
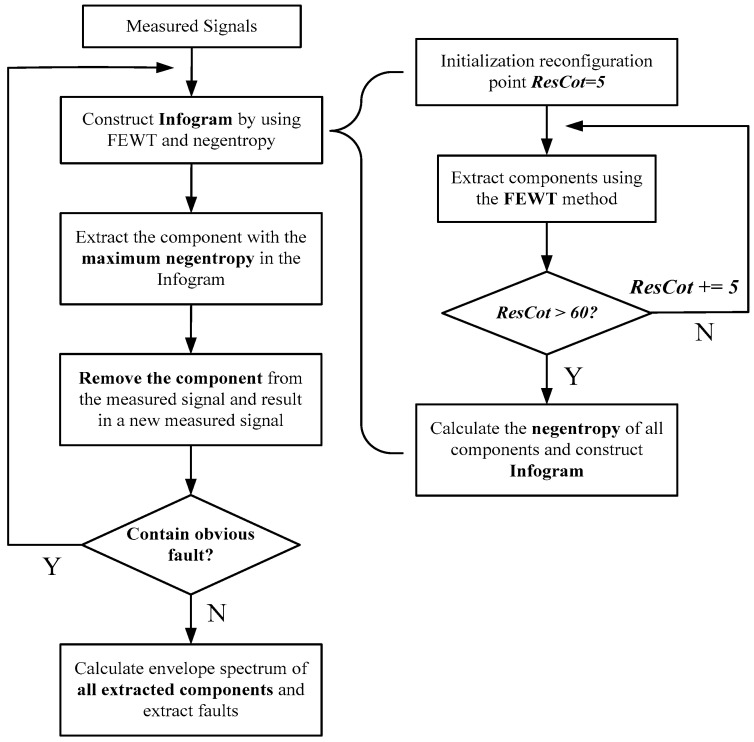
A flowchart of the negentropy spectrum decomposition method.

**Figure 11 entropy-21-00490-f011:**
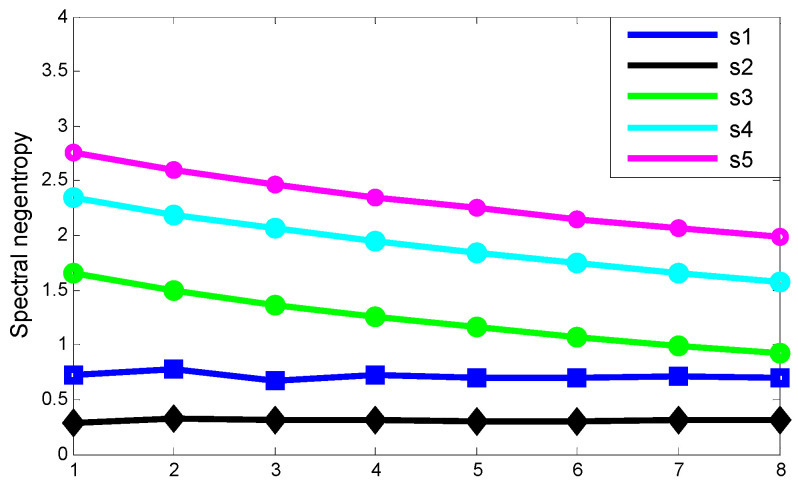
Spectral negentropy of noise signal, modulation signal and a series of transients.

**Figure 12 entropy-21-00490-f012:**
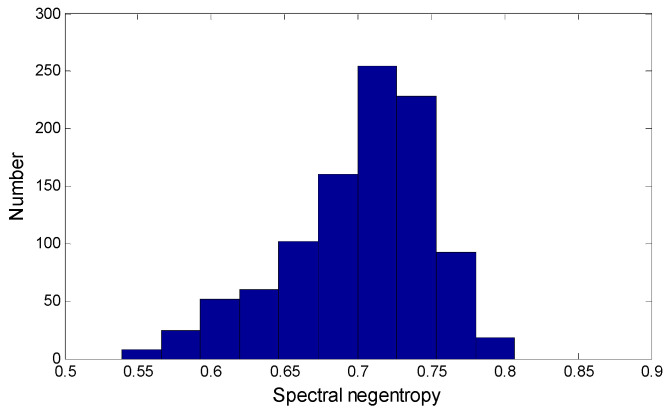
Spectral negentropy distribution of modulated signals with Gaussian noise.

**Figure 13 entropy-21-00490-f013:**
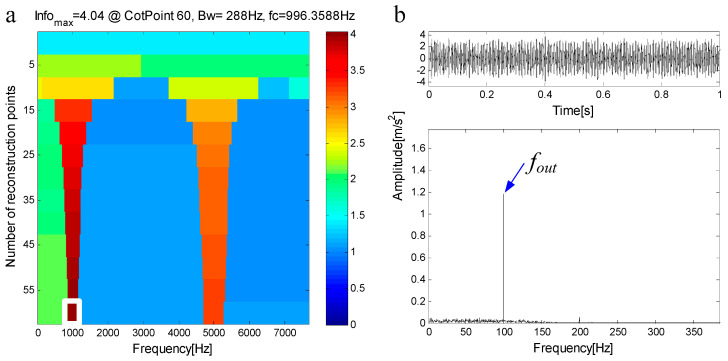
(**a**) Infogram based on FEWT; (**b**) component with ResPoint=60 and its envelope.

**Figure 14 entropy-21-00490-f014:**
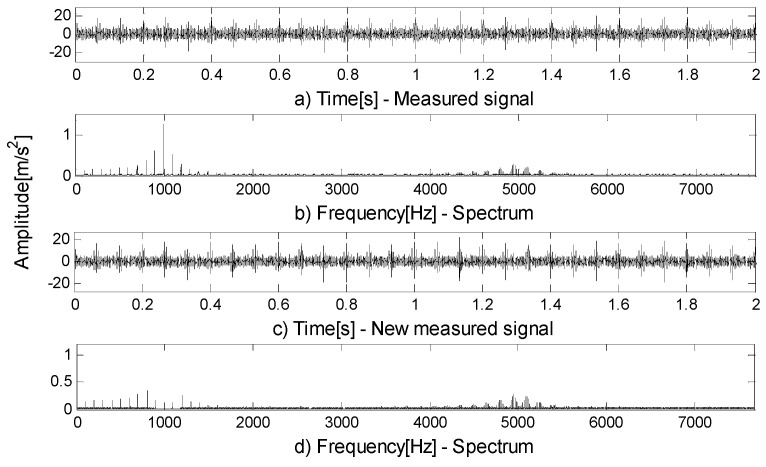
(**a**) Measured signal; (**b**) spectrum of (**a**); (**c**) new measured signal; (**d**) spectrum of (**c**).

**Figure 15 entropy-21-00490-f015:**
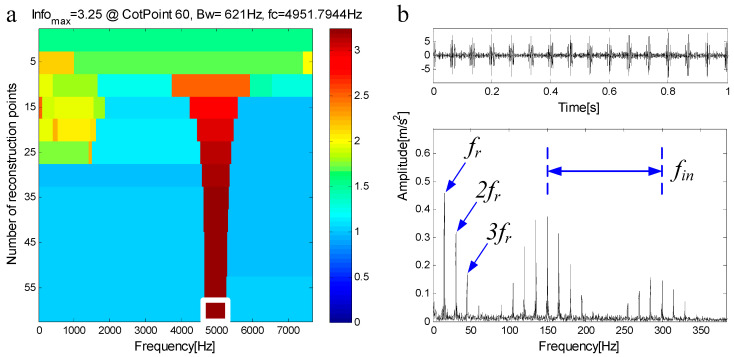
(**a**) Infogram based on FEWT; (**b**) component with ResPoint=60 and its envelope.

**Figure 16 entropy-21-00490-f016:**
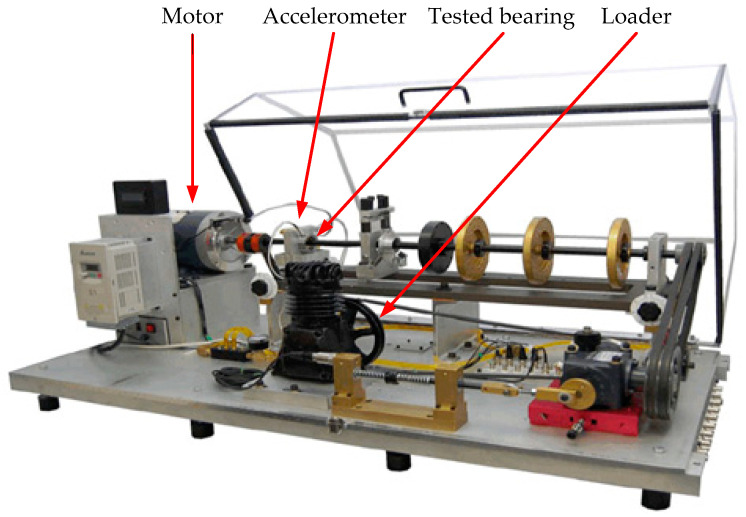
The Spectra Quest, Inc. test bench.

**Figure 17 entropy-21-00490-f017:**
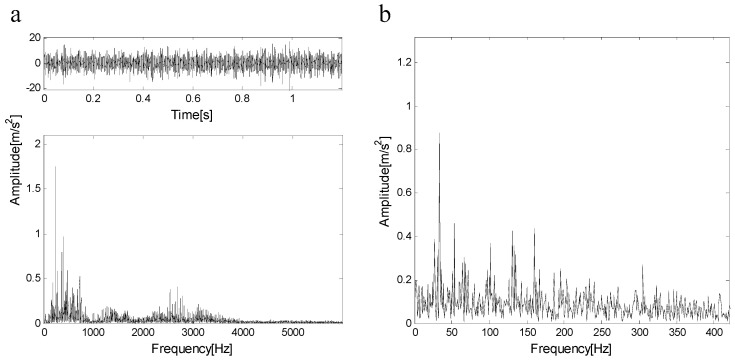
(**a**) The measured signal and its spectrum; (**b**) the envelope of the signal.

**Figure 18 entropy-21-00490-f018:**
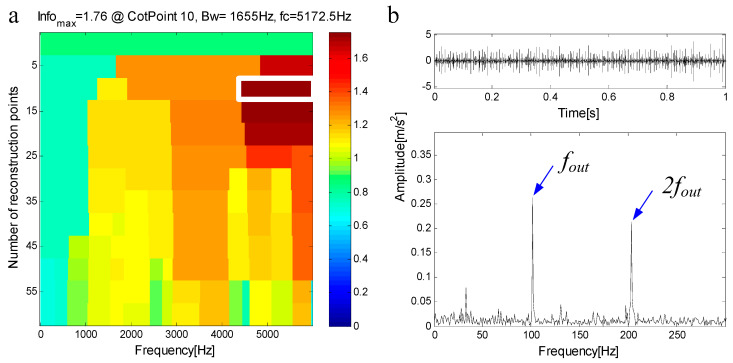
(**a**) Infogram based on FEWT; (**b**) component with ResPoint=10 and its envelope.

**Figure 19 entropy-21-00490-f019:**
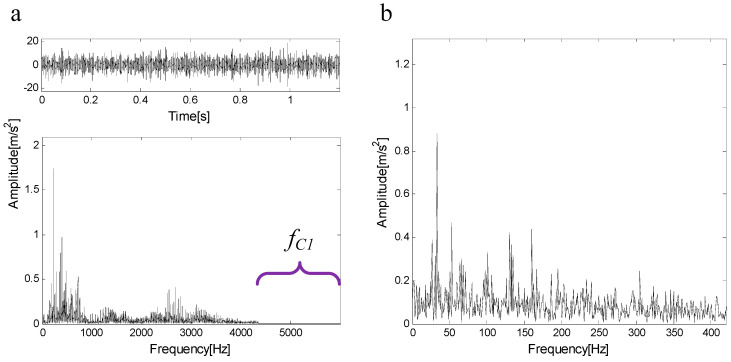
(**a**) The new measured signal and its spectrum after C1 extraction; (**b**) the envelope of the signal.

**Figure 20 entropy-21-00490-f020:**
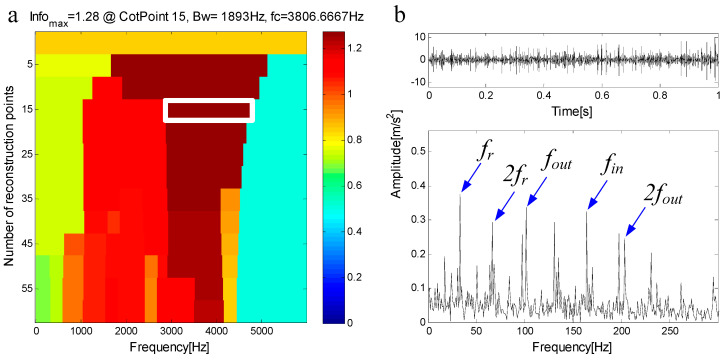
(**a**) Infogram based on FEWT; (**b**) component with ResPoint=15 and its envelope.

**Figure 21 entropy-21-00490-f021:**
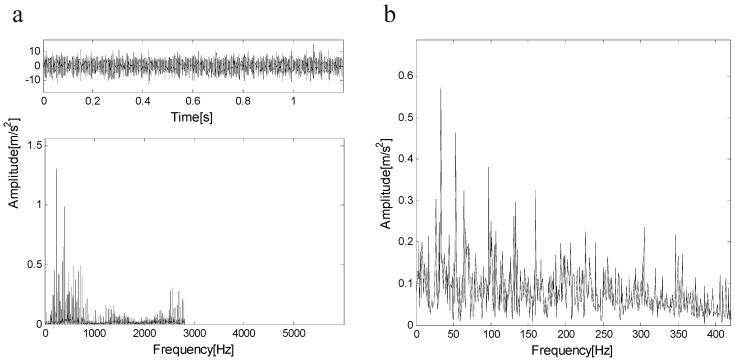
(**a**) The new measured signal and its spectrum after C2 extraction; (**b**) the envelope of the signal.

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
