# Peer review of "Negentropy Spectrum Decomposition and Its Application in Compound Fault Diagnosis of Rolling Bearing"

_entropy, 2019, doi:10.3390/e21050490_

Round 1
Reviewer 1 Report
The manuscript presents a signal-processing procedure that improves the accuracy of compound fault diagnosis of rolling bearings. The significance of the current research is fairly high however the manuscript needs to be improved in number of fronts before publication.
Here are some comments and suggestions:
· The literature review is pretty shallow and not critical. Currently seems more like a report that a scientific review. The references are just cited with little description. The authors can comment on significance of available research works and develop the research story of the manuscript in a better way. For example more critical review should be considered regarding compound fault diagnosis.
· The statements in the introduction sometimes are left without any justifications and some sentences do not make sense. (e.g. in lines 50 and 51: “It is very difficult for both researchers and engineers to understand how the intelligent classifiers work [18]”?!!).
· English is not of a good standards and significant proofreading seems necessary.
· There are numerous typos in text and equations that should be fixed(some examples : line 168 Equation (10) Sin function , in Line 188 almost the whole sentence, Lines 295, 305, 381, 393, 420, 432 spectral negentropy is ??!!, No equation number in line 347.
· The captions for the figures should be more descriptive and there should be a main figure caption followed by sub-captions.
· It is better to have more description and add labels for the main parts and elements to Figure 16. Currently it is just a photo with nothing else.

Author Response
Dear editor and reviewer,
I am very grateful to your comments for the manuscript. All of your suggestions are very important. They have important guiding significance for my thesis writing and scientific research work.
According to your suggestion, we revised the relevant part in manuscript. Due to limited space, we put most of the answers and revisions in the manuscript and the attachment of Word format.
I sincerely hope that all the responses are appropriate and acceptable to you. I also wish the revised manuscript meet the requirements for publication. If you have any questions about this paper, please don’t hesitate to let me know, and we will make a further discussion, and modify the manuscript.
Yours sincerely,
Junran Chen
E-mail:jrchen_1994@163.com

Reviewer 2 Report
In this paper developing a negentropy spectrum decomposition (NSD), the Authors are proposing a method that can detect and extract all the fault features of compound fault completely.
Combining fast empirical wavelet transform (FEWT) and NSD to construct the infogram, the fault feature information in the signal can be extracted accurately in the presence of impulsive noise.
The filtered signal is used as a new measured signal, and the above operation is repeated until the signal no longer contains obvious fault feature information.
The analysis of simulation signal and experimental signal shows that the method can realize the compound fault diagnosis of rolling bearings, which verifies the feasibility and effectiveness of the method.
I found that this paper is very interesting and that the obtained results are very promising, however in order to further improve I would only recommend to improve the conclusions and more references on the background (I suggest: doi: 10.3390/e21020135, doi: 10.3390/machines6030036, doi: doi.org/10.3390/e19030123, doi: 10.3390/e19040176, doi: 10.3390/app9081676).
Author Response

(The authors gave the same response as above.)

Reviewer 3 Report
The article deals with the diagnosis of rolling bearing failure. Specifically, it proposes a method for detecting compound bearing failure (in contrast to a single fault). The article argues about the lack of suitable methods for detecting compound failures (in contrast to a large number of methods for detecting single failures). The work is well structured; it is very technical but logical and legible. The analyses performed are sufficient and indicate the applicability of the proposed method. I only have smaller typographical comments: row 188 is not readable and some formulas could be better formatted (e.g. row 295).
Author Response

(The authors gave the same response as above.)

Reviewer 4 Report
The manuscript presents an algorithm for rolling bearing fault diagnosis based on some well-known techniques such as fast empirical wavelet transform (FEWT) and spectral negentropy. However the combination of these methods seems to be unique for this particular application. The following comments are given to further improve the manuscript quality:
1. Avoid lumping references, e.g. 12-14 and similar. Instead summarize the main contribution of each referenced paper in a separate sentence and/or cite the most recent and/or relevant one. The authors should also consider adding there a couple of more recently published results in the field, e.g.
https://www.mdpi.com/1424-8220/16/3/316 https://www.sciencedirect.com/science/article/abs/pii/S0360544216311525
2. The authors should more clearly explain why they use spectral negentropy for their problem and why this method turns out to be more accurate than some others.
3. Why did the authors choose this dataset to test their technique and not some other which contain much more data? Are there any other methods tested on this dataset and their results published? If yes, please compare those results with the results of the method presented in the manuscript. Have the authors tested the method for some other motor frequencies and how it performs in that case?
In overall the contribution of the manuscript is not so trivial at all but it needs a minor revision.
Author Response

(The authors gave the same response as above.)

Round 2
Reviewer 1 Report
Although some of the points have not been addressed completely but the manuscript has improved and can be published now.
Author Response
Dear editor and reviewer,
I am very grateful to your comments for the manuscript. All of your suggestions are very important. They have important guiding significance for my thesis writing and scientific research work.
Yours sincerely,
Junran Chen
E-mail:jrchen_1994@163.com
Reviewer 4 Report
The authors have improved the manuscript and took into account the most of comments. I still suggest they improve the introduction part a bit more by adding/describing there a couple of more recently published results in the field such as
https://www.mdpi.com/1424-8220/16/3/316 https://www.sciencedirect.com/science/article/abs/pii/S0360544216311525
Author Response

(The authors gave the same response as above.)
